# ACTIVE LEARNING FOR CONVOLUTIONAL NEURAL NETWORKS: A CORE-SET APPROACH

**Ozan Sener**[*]
Intel Labs
ozan.sener@intel.com

**Silvio Savarese**
Stanford University
ssilvio@stanford.edu

## ABSTRACT

Convolutional neural networks (CNNs) have been successfully applied to many recognition and learning tasks using a universal recipe; training a deep model on a very large dataset of supervised examples. However, this approach is rather restrictive in practice since collecting a large set of labeled images is very expensive. One way to ease this problem is coming up with smart ways for choosing images to be labelled from a very large collection (i.e. active learning).

Our empirical study suggests that many of the active learning heuristics in the literature are not effective when applied to CNNs in batch setting. Inspired by these limitations, we define the problem of active learning as *core-set selection*, i.e. choosing set of points such that a model learned over the selected subset is competitive for the remaining data points. We further present a theoretical result characterizing the performance of any selected subset using the geometry of the datapoints. As an active learning algorithm, we choose the subset which is expected to yield best result according to our characterization. Our experiments show that the proposed method significantly outperforms existing approaches in image classification experiments by a large margin.

## 1 INTRODUCTION

Deep convolutional neural networks (CNNs) have shown unprecedented success in many areas of research in computer vision and pattern recognition, such as image classification, object detection, and scene segmentation. Although CNNs are universally successful in many tasks, they have a major drawback; they need a very large amount of labeled data to be able to learn their large number of parameters. More importantly, it is almost always better to have more data since the accuracy of CNNs is often not saturated with increasing dataset size. Hence, there is a constant desire to collect more and more data. Although this a desired behavior from an algorithmic perspective (higher representative power is typically better), labeling a dataset is a time consuming and an expensive task. These practical considerations raise a critical question: *"what is the optimal way to choose data points to label such that the highest accuracy can be obtained given a fixed labeling budget."* Active learning is one of the common paradigms to address this question.

The goal of active learning is to find effective ways to choose data points to label, from a pool of unlabeled data points, in order to maximize the accuracy. Although it is not possible to obtain a universally good active learning strategy (Dasgupta, 2004), there exist many heuristics (Settles, 2010) which have been proven to be effective in practice. Active learning is typically an iterative process in which a model is learned at each iteration and a set of points is chosen to be labelled from a pool of unlabelled points using these aforementioned heuristics. We experiment with many of these heuristics in this paper and find them not effective when applied to CNNs. We argue that the main factor behind this ineffectiveness is the correlation caused via batch acquisition/sampling. In the classical setting, the active learning algorithms typically choose a single point at each iteration; however, this is not feasible for CNNs since i) a single point is likely to have no statistically significant impact on the accuracy due to the local optimization methods, and ii) each iteration requires a full training until convergence which makes it intractable to query labels one-by-one. Hence, it is necessary to query

---

[*]Work is completed while author is at Stanford University.

labels for a large subset at each iteration and it results in correlated samples even for moderately small subset sizes.

In order to tailor an active learning method for the batch sampling case, we decided to define the active learning as *core-set selection* problem. Core-set selection problem aims to find a small subset given a large labeled dataset such that a model learned over the small subset is competitive over the whole dataset. Since we have no labels available, we perform the core-set selection without using the labels. In order to attack the unlabeled core-set problem for CNNs, we provide a rigorous bound between an average loss over any given subset of the dataset and the remaining data points via the geometry of the data points. As an active learning algorithm, we try to choose a subset such that this bound is minimized. Moreover, minimization of this bound turns out to be equivalent to the k-Center problem (Wolf, 2011) and we adopt an efficient approximate solution to this combinatorial optimization problem. We further study the behavior of our proposed algorithm empirically for the problem of image classification using three different datasets. Our empirical analysis demonstrates state-of-the-art performance by a large margin.

## 2 RELATED WORK

We discuss the related work in the following categories separately. Briefly, our work is different from existing approaches in that $i$) it defines the active learning problem as core-set selection, $ii$) we consider both fully supervised and weakly supervised cases, and $iii$) we rigorously address the core-set selection problem directly for CNNs with no extra assumption.

**Active Learning** Active learning has been widely studied and most of the early work can be found in the classical survey of Settles (2010). It covers acquisition functions such as information theoretical methods (MacKay, 1992), ensemble approaches (McCallumzy & Nigamy, 1998; Freund et al., 1997) and uncertainty based methods (Tong & Koller, 2001; Joshi et al., 2009; Li & Guo, 2013).

Bayesian active learning methods typically use a non-parametric model like Gaussian process to estimate the expected improvement by each query (Kapoor et al., 2007) or the expected error after a set of queries (Roy & McCallum, 2001). These approaches are not directly applicable to large CNNs since they do not scale to large-scale datasets. A recent approach by Gal & Ghahramani (2016) shows an equivalence between dropout and approximate Bayesian inference enabling the application of Bayesian methods to deep learning. Although Bayesian active learning has been shown to be effective for small datasets (Gal et al., 2017), our empirical analysis suggests that they do not scale to large-scale datasets because of batch sampling.

One important class is that of uncertainty based methods, which try to find hard examples using heuristics like highest entropy (Joshi et al., 2009), and geometric distance to decision boundaries (Tong & Koller, 2001; Brinker, 2003). Our empirical analysis find them not to be effective for CNNs.

There are recent optimization based approaches which can trade-off uncertainty and diversity to obtain a diverse set of hard examples in batch mode active learning setting. Both Elhamifar et al. (2013) and Yang et al. (2015) design a discrete optimization problem for this purpose and use its convex surrogate. Similarly, Guo (2010) cast a similar problem as matrix partitioning. However, the optimization algorithms proposed in these papers use $n^2$ variables where $n$ is the number of data points. Hence, they do not scale to large datasets. There are also many pool based active learning algorithms designed for the specific class of machine learning algorithms like k-nearest neighbors and naive Bayes (Wei et al., 2015), logistic regression Hoi et al. (2006); Guo & Schuurmans (2008), and linear regression with Gaussian noise (Yu et al., 2006). Even in the algorithm agnostic case, one can design a set-cover algorithm to cover the hypothesis space using sub-modularity (Guillory & Bilmes, 2010; Golovin & Krause, 2011). On the other hand, Demir et al. (2011) uses a heuristic to first filter the pool based on uncertainty and then choose point to label using diversity. Our algorithm can be considered to be in this class; however, we do not use any uncertainty information. Our algorithm is also the first one which is applied to the CNNs. Most similar to ours are (Joshiy et al., 2010) and (Wang & Ye, 2015). Joshiy et al. (2010) uses a similar optimization problem. However, they offer no theoretical justification or analysis. Wang & Ye (2015) proposes to use empirical risk minimization like us; however, they try to minimize the difference between two distributions (maximum mean discrepancy between iid. samples from the dataset and the actively selected samples) instead of

core-set loss. Moreover, both algorithms are also not experimented with CNNs. In our experimental study, we compare with (Wang & Ye, 2015).

Recently, a discrete optimization based method (Berlind & Urner, 2015) which is similar to ours has been presented for k-NN type algorithms in the domain shift setting. Although our theoretical analysis borrows some techniques from them, their results are only valid for k-NNs.

Active learning algorithms for CNNs are also recently presented in (Wang et al., 2016; Stark et al., 2015). Wang et al. (2016) propose an heuristic based algorithm which directly assigns labels to the data points with high confidence and queries labels for the ones with low confidence. Moreover, Stark et al. (2015) specifically targets recognizing CAPTCHA images. Although their results are promising for CAPTCHA recognition, their method is not effective for image classification. We discuss limitations of both approaches in Section 5.

On the theoretical side, it is shown that greedy active learning is not possible in algorithm and data agnostic case (Dasgupta, 2005). However, there are data dependent results showing that it is indeed possible to obtain a query strategy which has better sample complexity than querying all points. These results either use assumptions about data-dependent realizability of the hypothesis space like (Gonen et al., 2013) or a data dependent measure of the concept space called disagreement coefficient (Hanneke, 2007). It is also possible to perform active learning in a batch setting using the greedy algorithm via importance sampling (Ganti & Gray, 2012). Although the aforementioned algorithms enjoy theoretical guarantees, they do not apply to large-scale problems.

**Core-Set Selection** The closest literature to our work is the problem of core-set selection since we define active learning as a core-set selection problem. This problem considers a fully labeled dataset and tries to choose a subset of it such that the model trained on the selected subset will perform as closely as possible to the model trained on the entire dataset. For specific learning algorithms, there are methods like core-sets for SVM (Tsang et al., 2005) and core-sets for k-Means and k-Medians (Har-Peled & Kushal, 2005). However, we are not aware of such a method for CNNs.

The most similar algorithm to ours is the unsupervised subset selection algorithm in (Wei et al., 2013). It uses a facility location problem to find a diverse cover for the dataset. Our algorithm differs in that it uses a slightly different formulation of facility location problem. Instead of the min-sum, we use the minimax (Wolf, 2011) form. More importantly, we apply this algorithm for the first time to the problem of active learning and provide theoretical guarantees for CNNs.

**Weakly-Supervised Deep Learning** Our paper is also related to semi-supervised deep learning since we experiment the active learning both in the fully-supervised and weakly-supervised scheme. One of the early weakly-supervised convolutional neural network algorithms was Ladder networks (Rasmus et al., 2015). Recently, we have seen adversarial methods which can learn a data distribution as a result of a two-player non-cooperative game (Salimans et al., 2016; Goodfellow et al., 2014; Radford et al., 2015). These methods are further extended to feature learning (Dumoulin et al., 2016; Donahue et al., 2016). We use Ladder networks in our experiments; however, our method is agnostic to the weakly-supervised learning algorithm choice and can utilize any model.

## 3 PROBLEM DEFINITION

In this section, we formally define the problem of active learning in the batch setting and set up the notation for the rest of the paper. We are interested in a $C$ class classification problem defined over a compact space $\mathcal{X}$ and a label space $\mathcal{Y} = \{1, \dots, C\}$. We also consider a loss function $l(\cdot, \cdot; \mathbf{w}) : \mathcal{X} \times \mathcal{Y} \to \mathcal{R}$ parametrized over the hypothesis class ($\mathbf{w}$), e.g. parameters of the deep learning algorithm. We further assume class-specific regression functions $\eta_c(\mathbf{x}) = p(y = c|\mathbf{x})$ to be $\lambda^\eta$-Lipschitz continuous for all $c$.

We consider a large collection of data points which are sampled $i.i.d.$ over the space $\mathcal{Z} = \mathcal{X} \times \mathcal{Y}$ as $\{\mathbf{x}_i, y_i\}_{i \in [n]} \sim p_\mathcal{Z}$ where $[n] = \{1, \dots, n\}$. We further consider an initial pool of data-points chosen uniformly at random as $\mathbf{s}^0 = \{s^0(j) \in [n]\}_{j \in [m]}$.

An active learning algorithm only has access to $\{\mathbf{x}_i\}_{i \in [n]}$ and $\{y_{s(j)}\}_{j \in [m]}$. In other words, it can only see the labels of the points in the initial sub-sampled pool. It is also given a budget $b$ of queries

to ask an oracle, and a learning algorithm $A_\mathbf{s}$ which outputs a set of parameters $\mathbf{w}$ given a labelled set $\mathbf{s}$. The active learning with a pool problem can simply be defined as

$$\min_{\mathbf{s}^1:|\mathbf{s}^1|\leq b} E_{\mathbf{x},y\sim p_\mathcal{Z}}[l(\mathbf{x},y;A_{\mathbf{s}^0\cup\mathbf{s}^1})] \tag{1}$$

In other words, an active learning algorithm can choose $b$ extra points and get them labelled by an oracle to minimize the future expected loss. There are a few differences between our formulation and the classical definition of active learning. Classical methods consider the case in which the budget is 1 ($b = 1$) but a single point has negligible effect in a deep learning regime hence we consider the batch case. It is also very common to consider multiple rounds of this game. We also follow the multiple round formulation with a myopic approach by solving the single round of labelling as;

$$\min_{\mathbf{s}^{k+1}:|\mathbf{s}^{k+1}|\leq b} E_{\mathbf{x},y\sim p_\mathcal{Z}}[l(\mathbf{x},y;A_{\mathbf{s}^0\cup\ldots\mathbf{s}^{k+1}})] \tag{2}$$

We only discuss the first iteration where $k = 0$ for brevity although we apply it over multiple rounds.

At each iteration, an active learning algorithm has two stages: 1. identifying a set of data-points and presenting them to an oracle to be labelled, and 2. training a classifier using both the new and the previously labeled data-points. The second stage (training the classifier) can be done in a fully or weakly-supervised manner. Fully-supervised is the case where training the classifier is done using only the labeled data-points. Weakly-supervised is the case where training also utilizes the points which are not labelled yet. Although the existing literature only focuses on the active learning for fully-supervised models, we consider both cases and experiment on both.

## 4 METHOD

### 4.1 ACTIVE LEARNING AS A SET COVER

In the classical active learning setting, the algorithm acquires labels one by one by querying an oracle (i.e. $b = 1$). Unfortunately, this is not feasible when training CNNs since $i$) a single point will not have a statistically significant impact on the model due to the local optimization algorithms. $ii$) it is infeasible to train as many models as number of points since many practical problem of interest is very large-scale. Hence, we focus on the batch active learning problem in which the active learning algorithm choose a moderately large set of points to be labelled by an oracle at each iteration.

In order to design an active learning strategy which is effective in batch setting, we consider the following upper bound of the active learning loss we formally defined in (1):

$$E_{\mathbf{x},y\sim p_\mathcal{Z}}[l(\mathbf{x},y;A_\mathbf{s})] \leq \underbrace{\left|E_{\mathbf{x},y\sim p_\mathcal{Z}}[l(\mathbf{x},y;A_\mathbf{s})] - \frac{1}{n}\sum_{i\in[n]}l(\mathbf{x}_i,y_i;A_\mathbf{s})\right|}_{\text{Generalization Error}} + \underbrace{\frac{1}{|\mathbf{s}|}\sum_{j\in\mathbf{s}}l(\mathbf{x}_j,y_j;A_\mathbf{s})}_{\text{Training Error}}$$
$$+ \underbrace{\left|\frac{1}{n}\sum_{i\in[n]}l(\mathbf{x}_i,y_i;A_\mathbf{s}) - \frac{1}{|\mathbf{s}|}\sum_{j\in\mathbf{s}}l(\mathbf{x}_j,y_j;A_\mathbf{s}),\right|}_{\text{Core-Set Loss}} \tag{3}$$

The quantity we are interested in is the population risk of the model learned using a small labelled subset ($\mathbf{s}$). The population risk is controlled by the *training error* of the model on the labelled subset, the *generalization error* over the full dataset ($[n]$) and a term we define as the *core-set loss*. Core-set loss is simply the difference between average empirical loss over the set of points which have labels for and the average empirical loss over the entire dataset including unlabelled points. Empirically, it is widely observed that the CNNs are highly expressive leading to very low training error and they typically generalize well for various visual problems. Moreover, generalization error of CNNs is also theoretically studied and shown to be bounded by Xu & Mannor (2012). Hence, the critical part for active learning is the core-set loss. Following this observation, we re-define the active learning problem as:

$$\min_{\mathbf{s}^1:|\mathbf{s}^1|\leq b} \left|\frac{1}{n}\sum_{i\in[n]}l(\mathbf{x}_i,y_i;A_{\mathbf{s}^0\cup\mathbf{s}^1}) - \frac{1}{|\mathbf{s}^0+\mathbf{s}^1|}\sum_{j\in\mathbf{s}^0\cup\mathbf{s}^1}l(\mathbf{x}_j,y_j;A_{\mathbf{s}^0\cup\mathbf{s}^1})\right| \tag{4}$$

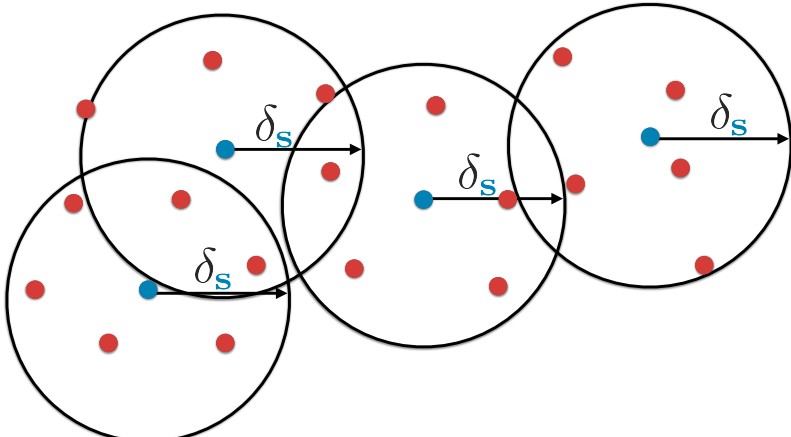

Figure 1: **Visualization of the Theorem 1**. Consider the set of selected points **s** and the points in the remainder of the dataset $[n] \setminus \mathbf{s}$, our results shows that if **s** is the $\delta_\mathbf{s}$ cover of the dataset,
$$\left| \frac{1}{n} \sum_{i \in [n]} l(\mathbf{x}_i, y_i, A_\mathbf{s}) - \frac{1}{|\mathbf{s}|} \sum_{j \in \mathbf{s}} l(\mathbf{x}_j, y_j; A_\mathbf{s}) \right| \leq \mathcal{O}(\delta_\mathbf{s}) + \mathcal{O}\left(\sqrt{\frac{1}{n}}\right)$$

Informally, given the initial labelled set ($\mathbf{s}^0$) and the budget ($b$), we are trying to find a set of points to query labels ($\mathbf{s}^1$) such that when we learn a model, the performance of the model on the labelled subset and that on the whole dataset will be as close as possible.

## 4.2 CORE-SETS FOR CNNs

The optimization objective we define in (4) is not directly computable since we do not have access to all the labels (i.e. $[n] \setminus (\mathbf{s}^0 \cup \mathbf{s}^1)$ is unlabelled). Hence, in this section we give an upper bound for this objective function which we can optimize.

We start with presenting this bound for any loss function which is Lipschitz for a fixed true label $y$ and parameters **w**, and then show that loss functions of CNNs with ReLu non-linearities satisfy this property. We also rely on the zero training error assumption. Although the zero training error is not an entirely realistic assumption, our experiments suggest that the resulting upper bound is very effective. We state the following theorem;

**Theorem 1.** *Given $n$ i.i.d. samples drawn from $p_\mathcal{Z}$ as $\{\mathbf{x}_i, y_i\}_{i \in [n]}$, and set of points* **s**. *If loss function $l(\cdot, y, \mathbf{w})$ is $\lambda^l$-Lipschitz continuous for all $y$, **w** and bounded by $L$, regression function is $\lambda^\eta$-Lipschitz, **s** is $\delta_\mathbf{s}$ cover of $\{\mathbf{x}_i, y_i\}_{i \in [n]}$, and $l(\mathbf{x}_{s(j)}, y_{s(j)}; A_\mathbf{S}) = 0 \quad \forall j \in [m]$; with probability at least $1 - \gamma$,*
$$\left| \frac{1}{n} \sum_{i \in [n]} l(\mathbf{x}_i, y_i; A_\mathbf{s}) - \frac{1}{|\mathbf{s}|} \sum_{j \in \mathbf{s}} l(\mathbf{x}_j, y_j; A_\mathbf{s}) \right| \leq \delta(\lambda^l + \lambda^\mu LC) + \sqrt{\frac{L^2 \log(1/\gamma)}{2n}}.$$

Since we assume a zero training error for core-set, the core-set loss is equal to the average error over entire dataset as $\left| \frac{1}{n} \sum_{i \in [n]} l(\mathbf{x}_i, y_i; A_\mathbf{s}) - \frac{1}{|\mathbf{s}|} \sum_{j \in \mathbf{s}} l(\mathbf{x}_j, y_j; A_\mathbf{s}) \right| = \frac{1}{n} \sum_{i \in [n]} l(\mathbf{x}_i, y_i; A_\mathbf{s})$. We state the theorem in this form to be consistent with (3). We visualize this theorem in Figure 1 and defer its proof to the appendix. In this theorem, "a set **s** is a $\delta$ cover of a set $s^\star$" means a set of balls with radius $\delta$ centered at each member of **s** can cover the entire $s^\star$. Informally, this theorem suggests that we can bound the core-set loss with covering radius and a term which goes to zero with rate depends solely on $n$. This is an interesting result since this bound does not depend on the number of labelled points. In other words, a provided label does not help the core-set loss unless it decreases the covering radius.

In order to show that this bound applies to CNNs, we prove the Lipschitz-continuity of the loss function of a CNN with respect to input image for a fixed true label with the following lemma where max-pool and restricted linear units are the non-linearities and the loss is defined as the $l_2$

distance between the desired class probabilities and the soft-max outputs. CNNs are typically used with cross-entropy loss for classification problems in the literature. Indeed, we also perform our experiments using the cross-entropy loss although we use $l_2$ loss in our theoretical study. Although our theoretical study does not extend to cross-entropy loss, our experiments suggest that the resulting algorithm is very effective for cross-entropy loss.

**Lemma 1.** *Loss function defined as the 2-norm between the class probabilities and the softmax output of a convolutional neural network with $n_c$ convolutional (with max-pool and ReLU) and $n_{fc}$ fully connected layers defined over C classes is $\left( \frac{\sqrt{C-1}}{C} \alpha^{n_c + n_{fc}} \right)$-Lipschitz function of input for fixed class probabilities and network parameters.*

Here, $\alpha$ is the maximum sum of input weights per neuron (see appendix for formal definition). Although it is in general unbounded, it can be made arbitrarily small without changing the loss function behavior (i.e. keeping the label of any data point **s** unchanged). We defer the proof to the appendix and conclude that CNNs enjoy the bound we presented in Theorem 1.

In order to computationally perform active learning, we use this upper bound. In other words, the practical problem of interest becomes $\min_{\mathbf{s}^1 : |\mathbf{s}^1| \leq b|} \delta_{\mathbf{s}^0 \cup \mathbf{s}^1}$. This problem is equivalent to the k-Center problem (also called min-max facility location problem) (Wolf, 2011). In the next section, we explain how we solve the k-Center problem in practice using a greedy approximation.

## 4.3 SOLVING THE K-CENTER PROBLEM

We have so far provided an upper bound for the loss function of the core-set selection problem and showed that minimizing it is equivalent to the *k-Center* problem (minimax facility location (Wolf, 2011)) which can intuitively be defined as follows; choose $b$ center points such that the largest distance between a data point and its nearest center is minimized. Formally, we are trying to solve:

$$\min_{\mathbf{s}^1 : |\mathbf{s}^1| \leq b} \max_i \min_{j \in \mathbf{s}^1 \cup \mathbf{s}^0} \Delta(\mathbf{x}_i, \mathbf{x}_j) \quad (5)$$

---

**Algorithm 1** k-Center-Greedy

**Input:** data $\mathbf{x}_i$, existing pool $\mathbf{s}^0$ and a budget $b$
Initialize $\mathbf{s} = \mathbf{s}^0$
**repeat**
$\quad u = \arg\max_{i \in [n] \backslash \mathbf{s}} \min_{j \in \mathbf{s}} \Delta(\mathbf{x}_i, \mathbf{x}_j)$
$\quad \mathbf{s} = \mathbf{s} \cup \{u\}$
**until** $|\mathbf{s}| = b + |\mathbf{s}^0|$
**return** $\mathbf{s} \backslash \mathbf{s}^0$

---

Unfortunately this problem is NP-Hard (Cook et al., 1998). However, it is possible to obtain a $2 - OPT$ solution efficiently using a greedy approach shown in Algorithm 1. If $OPT = \min_{\mathbf{s}^1} \max_i \min_{j \in \mathbf{s}^1 \cup \mathbf{s}^0} \Delta(\mathbf{x}_i, \mathbf{x}_j)$, the greedy algorithm shown in Algorithm 1 is proven to have a solution $(\mathbf{s}^1)$ such that; $\max_i \min_{j \in \mathbf{s}^1 \cup \mathbf{s}^0} \Delta(\mathbf{x}_i, \mathbf{x}_j) \leq 2 \times OPT$.

Although the greedy algorithm gives a good initialization, in practice we can improve the $2 - OPT$ solution by iteratively querying upper bounds on the optimal value. In other words, we can design an algorithm which decides if $OPT \leq \delta$. In order to do so, we define a mixed integer program (MIP) parametrized by $\delta$ such that its feasibility indicates $\min_{\mathbf{s}^1} \max_i \min_{j \in \mathbf{s}^1 \cup \mathbf{s}^0} \Delta(\mathbf{x}_i, \mathbf{x}_j) \leq \delta$. A straight-forward algorithm would be to use this MIP as a sub-routine and performing a binary search between the result of the greedy algorithm and its half since the optimal solution is guaranteed to be included in that range. While constructing this MIP, we also try to handle one of the weaknesses of k-Center algorithm, namely robustness. To make the k-Center problem robust, we assume an upper limit on the number of outliers $\Xi$ such that our algorithm can choose not to cover at most $\Xi$ unsupervised data points. This mixed integer program can be written as:

$$Feasible(b, \mathbf{s}^0, \delta, \Xi) : \sum_j u_j, = |\mathbf{s}^0| + b, \qquad \sum_{i,j} \xi_{i,j} \leq \Xi$$

$$\sum_j \omega_{i,j} = 1 \quad \forall i, \qquad \omega_{i,j} \leq u_j \quad \forall i, j$$

$$u_i = 1 \quad \forall i \in \mathbf{s}^0, \qquad u_i \in \{0, 1\} \quad \forall i \qquad (6)$$

$$\omega_{i,j} = \xi_{i,j} \quad \forall i, j \quad | \quad \Delta(\mathbf{x}_i, \mathbf{x}_j) > \delta.$$

In this formulation, $u_i$ is 1 if the $i^{th}$ data point is chosen as center, $\omega_{i,j}$ is 1 if the $i^{th}$ point is covered by the $j^{th}$, point and $\xi_{i,j}$ is 1 if the $i^{th}$ point is an outlier and covered by the $j^{th}$ point without the $\delta$

constraint, and $0$ otherwise. And, variables are binary as $u_i, \omega_{i,j}, \xi_{i,j} \in \{0, 1\}$. We further visualize these variables in a diagram in Figure 2, and give the details of the method in Algorithm 2.

---

**Algorithm 2** Robust k-Center

**Input:** data $\mathbf{x}_i$, existing pool $\mathbf{s}^0$, budget $b$ and outlier bound $\Xi$
**Initialize** $\mathbf{s}_g = $ k-Center-Greedy$(\mathbf{x}_i, \mathbf{s}^0, b)$
$\delta_{2-OPT} = \max_j \min_{i \in \mathbf{s}_g} \Delta(\mathbf{x}_i, \mathbf{x}_j)$
$lb = \frac{\delta_{2-OPT}}{2}, ub = \delta_{2-OPT}$
**repeat**
   **if** $Feasible(b, \mathbf{s}^0, \frac{lb+ub}{2}, \Xi)$ **then**
      $ub = \max_{i,j|\Delta(\mathbf{x}_i, \mathbf{x}_j) \leq \frac{lb+ub}{2}} \Delta(\mathbf{x}_i, \mathbf{x}_j)$
   **else**
      $lb = \min_{i,j|\Delta(\mathbf{x}_i, \mathbf{x}_j) \geq \frac{lb+ub}{2}} \Delta(\mathbf{x}_i, \mathbf{x}_j)$
   **end if**
**until** $ub = lb$
**return** $\{i \ s.t. \ u_i = 1\}$

---

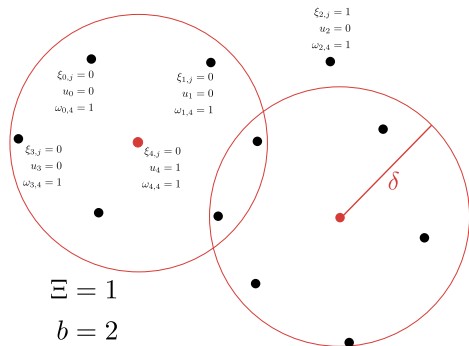

$$\Xi = 1$$
$$b = 2$$

Figure 2: Visualizations of the variables. In this solution, the $4^{th}$ node is chosen as a center and nodes $0, 1, 3$ are in a $\delta$ ball around it. The $2^{nd}$ node is marked as an outlier.

## 4.4 IMPLEMENTATION DETAILS

One of the critical design choices is the distance metric $\Delta(\cdot, \cdot)$. We use the $l_2$ distance between activations of the final fully-connected layer as the distance. For weakly-supervised learning, we used Ladder networks (Rasmus et al., 2015) and for all experiments we used VGG-16 (Simonyan & Zisserman, 2014) as the CNN architecture. We initialized all convolutional filters according to He et al. (2016). We optimized all models using RMSProp with a learning rate of $1\mathrm{e}{-3}$ using Tensorflow (Abadi et al., 2016). We train CNNs from scratch after each iteration.

We used the Gurobi (Inc., 2016) framework for checking feasibility of the MIP defined in (6). As an upper bound on outliers, we used $\Xi = 1\mathrm{e}{-4} \times n$ where $n$ is the number of unlabelled points.

## 5 EXPERIMENTAL RESULTS

We tested our algorithm on the problem of classification using three different datasets. We performed experiments on CIFAR (Krizhevsky & Hinton, 2009) and Caltech-256 (Griffin et al., 2007) datasets for image classification and on SVHN(Netzer et al., 2011) dataset for digit classification. CIFAR (Krizhevsky & Hinton, 2009) dataset has two tasks; one coarse-grained over 10 classes and one fine-grained over 100 classes. We performed experiments on both.

We compare our method with the following baselines: $i$)**Random:** Choosing the points to be labelled uniformly at random from the unlabelled pool. $ii$)**Best Empirical Uncertainty:** Following the empirical setup in (Gal et al., 2017), we perform active learning using max-entropy, BALD and Variation Ratios treating soft-max outputs as probabilities. We only report the best performing one for each dataset since they perform similar to each other. $iii$) **Deep Bayesian Active Learning (DBAL)(Gal et al., 2017):** We perform Monte Carlo dropout to obtain improved uncertainty measures and report only the best performing acquisition function among max-entropy, BALD and Variation Ratios for each dataset. $iv$) **Best Oracle Uncertainty:** We also report a best performing oracle algorithm which uses the label information for entire dataset. We replace the uncertainty with $l(\mathbf{x}_i, y_i, A_{\mathbf{s}^0})$ for all unlabelled examples. We sample the queries from the normalized form of this function by setting the probability of choosing the $i^{th}$ point to be queried as $p_i = \frac{l(\mathbf{x}_i, y_i, A_{\mathbf{s}^0})}{\sum_j l(\mathbf{x}_j, y_j, A_{\mathbf{s}^0})}$. $v$)**k-Median:** Choosing the points to be labelled as the cluster centers of k-Median (k is equal to the budget) algorithm. $vi$)**Batch Mode Discriminative-Representative Active Learning(BMDR)(Wang & Ye, 2015):** ERM based approach which uses uncertainty and minimizes MMD between iid. samples from the dataset and the actively chosen points. $vii$)**CEAL (Wang et al., 2016):** CEAL (Wang et al., 2016) is a weakly-supervised active learning method proposed specifically for CNNs. we include it in the weakly-supervised analysis.

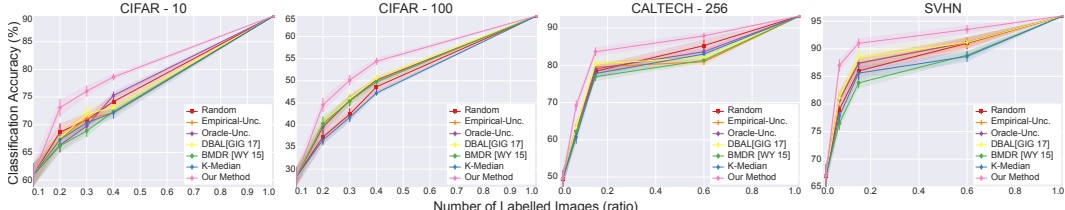

Figure 3: Results on Active Learning for Weakly-Supervised Model (error bars are std-dev)

Figure 4: Results on Active Learning for Fully-Supervised Model (error bars are std-dev)

We conducted experiments on active learning for fully-supervised models as well as active learning for weakly-supervised models. In our experiments, we start with small set of images sampled uniformly at random from the dataset as an initial pool. The weakly-supervised model has access to labeled examples as well as unlabelled examples. The fully-supervised model only has access to the labeled data points. We run all experiments with five random initializations of the initial pool of labeled points and use the average classification accuracy as a metric. We plot the accuracy vs the number of labeled points. We also plot error bars as standard deviations. We run the query algorithm iteratively; in other words, we solve the discrete optimization problem $\min_{\mathbf{s}^{k+1}:|\mathbf{s}^{k+1}|\leq b} E_{\mathbf{x},y\sim p_{\mathcal{Z}}}[l(\mathbf{x},y;A_{\mathbf{s}^0\cup\ldots,\mathbf{s}^{k+1}})]$ for each point on the accuracy vs number of labelled examples graph. We present the results in Figures 3 and 4.

Figures 3 and 4 suggests that our algorithm outperforms all other baselines in all experiments; for the case of weakly-supervised models, by a large margin. We believe the effectiveness of our approach in the weakly-supervised case is due to the better feature learning. Weakly-supervised models provide better feature spaces resulting in accurate geometries. Since our method is geometric, it performs significantly better with better feature spaces. We also observed that our algorithm is less effective in CIFAR-100 and Caltech-256 when compared with CIFAR-10 and SVHN. This can easily be explained using our theoretical analysis. Our bound over the core-set loss scales with the number of classes, hence it is better to have fewer classes.

One interesting observation is the fact that a state-of-the-art batch mode active learning baseline (BMDR (Wang & Ye, 2015)) does not necessarily perform better than greedy ones. We believe this is due to the fact that it still uses an uncertainty information and soft-max probabilities are not a good proxy for uncertainty. Our method does not use any uncertainty. And, incorporating uncertainty to our method in a principled way is an open problem and a fruitful future research direction. On the other hand, a pure clustering based batch active learning baseline (k-Medoids) is also not effective. We believe this is rather intuitive since cluster sentences are likely the points which are well covered with initial iid. samples. Hence, this clustering based method fails to sample the tails of the data distribution.

Our results suggest that both oracle uncertainty information and Bayesian estimation of uncertainty is helpful since they improve over empirical uncertainty baseline; however, they are still not effective in the batch setting since random sampling outperforms them. We believe this is due to the correlation in the queried labels as a consequence of active learning in batch setting. We further investigate this with a qualitative analysis via tSNE (Maaten & Hinton, 2008) embeddings. We compute embeddings for all points using the features which are learned using the labelled examples and visualize the points sampled by our method as well as the oracle uncertainty. This visualization suggests that due to the correlation among samples, uncertainty based methods fail to cover the large portion of the space confirming our hypothesis.

**Optimality of the k-Center Solution:** Our proposed method uses the greedy 2-OPT solution for the k-Center problem as an initialization and checks the feasibility of a mixed integer program (MIP).

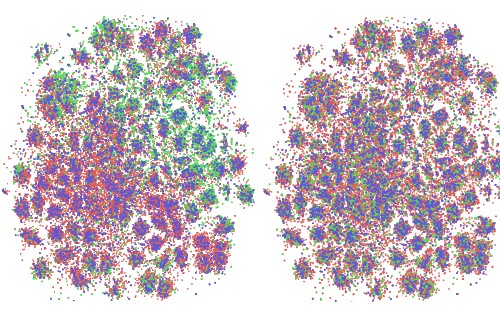

(a) Uncertainty Oracle          (b) Our Method

Figure 5: tSNE embeddings of the CIFAR dataset and behavior of uncertainty oracle as well as our method. For both methods, the initial labeled pool of 1000 images are shown in blue, 1000 images chosen to be labeled in green and remaining ones in red. Our algorithm results in queries evenly covering the space. On the other hand, samples chosen by uncertainty oracle fails to cover the large portion of the space.

Table 1: Average run-time of our algorithm for $b = 5k$ and $|\mathbf{s}^0| = 10k$ in seconds.

| Distance Matrix | Greedy (2-OPT) | MIP (iteration) | MIP (total) | Total |
|---|---|---|---|---|
| 104.2 | 2 | 7.5 | 244.03 | 360.23 |

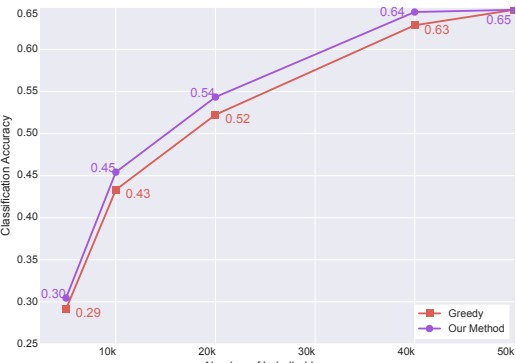

Figure 6: We compare our method with k-Center-Greedy. Our algorithm results in a small but important accuracy improvement.

We use LP-relaxation of the defined MIP and use branch-and-bound to obtain integer solutions. The utility obtained by solving this expensive MIP should be investigated. We compare the average run-time of MIP[1] with the run-time of 2-OPT solution in Table 1. We also compare the accuracy obtained with optimal k-Center solution and the 2-OPT solution in Figure 6 on CIFAR-100 dataset.

As shown in the Table 1; although the run-time of MIP is not polynomial in worst-case, in practice it converges in a tractable amount of time for a dataset of 50k images. Hence, our algorithm can easily be applied in practice. Figure 6 suggests a small but significant drop in the accuracy when the 2-OPT solution is used. Hence, we conclude that unless the scale of the dataset is too restrictive, using our proposed optimal solver is desired. Even with the accuracy drop, our active learning strategy using 2-OPT solution still outperforms the other baselines. Hence, we can conclude that our algorithm can scale to any dataset size with small accuracy drop even if solving MIP is not feasible.

## 6    CONCLUSION

We study the active learning problem for CNNs. Our empirical analysis showed that classical uncertainty based methods have limited applicability to the CNNs due to the correlations caused by batch sampling. We re-formulate the active learning problem as core-set selection and study the core-set problem for CNNs. We further validated our algorithm using an extensive empirical study. Empirical results on three datasets showed state-of-the-art performance by a large margin.

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

## A  PROOF FOR LEMMA 1

*Proof.* We will start with showing that softmax function defined over $C$ class is $\frac{\sqrt{C-1}}{C}$-Lipschitz continuous. It is easy to show that for any differentiable function $f : \mathbb{R}^n \to \mathbb{R}^m$,

$$\|f(\mathbf{x}) - f(\mathbf{y})\|_2 \leq \|J\|_F^* \|\mathbf{x} - \mathbf{y}\|_2 \ \ \forall \mathbf{x}, \mathbf{y} \in \mathbb{R}^n$$

where $\|J\|_F^* = \max_{\mathbf{x}} \|J\|_F$ and $J$ is the Jacobian matrix of $f$.

Softmax function is defined as

$$f(x)_i = \frac{\exp(x_i)}{\sum\limits_{j=1}^{C} \exp(x_j)}, \ i = 1, 2, ..., C$$

For brevity, we will denote $f_i(x)$ as $f_i$. The Jacobian matrix will be,

$$J = \begin{bmatrix} f_1(1 - f_1) & -f_1 f_2 & ... & -f_1 f_C \\ -f_2 f_1 & f_2(1 - f_2) & ... & -f_2 f_C \\ ... & ... & ... & ... \\ -f_C f_1 & -f_C f_2 & ... & -f_C(1 - f_C) \end{bmatrix}$$

Now, Frobenius norm of above matrix will be,

$$\|J\|_F = \sqrt{\sum_{i=1}^{C} \sum_{j=1, i \neq j}^{C} f_i^2 f_j^2 + \sum_{i=1}^{C} f_i^2 (1 - f_i)^2}$$

It is straightforward to show that $f_i = \frac{1}{C}$ is the optimal solution for $\|J\|_F^* = \max_x \|J\|_F$ Hence, putting $f_i = \frac{1}{C}$ in the above equation , we get $\|J\|_F^* = \frac{\sqrt{C-1}}{C}$.

Now, consider two inputs $\mathbf{x}$ and $\tilde{\mathbf{x}}$, such that their representation at layer $d$ is $\mathbf{x}^d$ and $\tilde{\mathbf{x}}^d$. Let's consider any convolution or fully-connected layer as $\mathbf{x}_j^d = \sum_i w_{i,j}^d \mathbf{x}_i^{d-1}$. If we assume, $\sum_i |w_{i,j}| \leq \alpha \ \ \forall i, j, d$, for any convolutional or fully connected layer, we can state:

$$\|\mathbf{x}^d - \tilde{\mathbf{x}}^d\|_2 \leq \alpha \|\mathbf{x}^{d-1} - \tilde{\mathbf{x}}^{d-1}\|_2$$

On the other hand, using $|a - b| \leq |\max(0,a) - \max(0,a)|$ and the fact that max pool layer can be written as a convolutional layer such that only one weight is 1 and others are 0, we can state for ReLU and max-pool layers,

$$\|\mathbf{x}^d - \tilde{\mathbf{x}}^d\|_2 \leq \|\mathbf{x}^{d-1} - \tilde{\mathbf{x}}^{d-1}\|_2$$

Combining with the Lipschitz constant of soft-max layer,

$$\|CNN(\mathbf{x};\mathbf{w}) - CNN(\tilde{\mathbf{x}};\mathbf{w})\|_2 \leq \frac{\sqrt{C-1}}{C}\alpha^{n_c + n_{fc}}\|\mathbf{x} - \tilde{\mathbf{x}}\|_2$$

Using the reverse triangle inequality as

$$|l(\mathbf{x},y;\mathbf{w}) - l(\tilde{\mathbf{x}},y;\mathbf{w})| = |\|CNN(\mathbf{x};\mathbf{w}) - y\|_2 - \|CNN(\tilde{\mathbf{x}};\mathbf{w}) - y\|_2| \leq \|CNN(\mathbf{x};\mathbf{w}) - CNN(\tilde{\mathbf{x}};\mathbf{w})\|_2,$$

we can conclude that the loss function is $\frac{\sqrt{C-1}}{C}\alpha^{n_c + n_{fc}}$-Lipschitz for any fixed $y$ and $\mathbf{w}$. □

## B  PROOF FOR THEOREM 1

Before starting our proof, we state the Claim 1 from Berlind & Urner (2015). Fix some $p, p' \in [0,1]$ and $y' \in \{0,1\}$. Then,

$$p_{y \sim p}(y \neq y') \leq p_{y \sim p'}(y \neq y') + |p - p'|$$

*Proof.* We will start our proof with bounding $E_{y_i \sim \eta(\mathbf{x}_i)}[l(\mathbf{x}_i, y_i; A_\mathbf{s})]$. We have a condition which states that there exists and $\mathbf{x}_j$ in $\delta$ ball around $\mathbf{x}_i$ such that $\mathbf{x}_j$ has 0 loss.

$$E_{y_i \sim \eta(\mathbf{x}_i)}[l(\mathbf{x}_i, y_i; A_\mathbf{s})] = \sum_{k \in [C]} p_{y_i \sim \eta_k(\mathbf{x}_i)}(y_i = k) l(\mathbf{x}_i, k; A_\mathbf{s})$$

$$\overset{(d)}{\leq} \sum_{k \in [C]} p_{y_i \sim \eta_k(\mathbf{x}_j)}(y_i = k) l(\mathbf{x}_i, k; A_\mathbf{s}) + \sum_{k \in [C]} |\eta_k(\mathbf{x}_i) - \eta_k(\mathbf{x}_j)| l(\mathbf{x}_i, k; A_\mathbf{s})$$

$$\overset{(e)}{\leq} \sum_{k \in [C]} p_{y_i \sim \eta_k(\mathbf{x}_j)}(y_i = k) l(\mathbf{x}_i, k; A_\mathbf{s}) + \delta \lambda^\eta L C$$

With abuse of notation, we represent $\{y_i = k\} \sim \eta_k(\mathbf{x}_i)$ with $y_i \sim \eta_k(\mathbf{x}_i)$. We use Claim 1 in $(d)$, and Lipschitz property of regression function and bound of loss in $(d)$. Then, we can further bound the remaining term as;

$$\sum_{k \in [C]} p_{y_i \sim \eta_k(\mathbf{x}_j)}(y_i = k) l(\mathbf{x}_i, k; A_\mathbf{s}) = \sum_{k \in [C]} p_{y_i \sim \eta_k(\mathbf{x}_j)}(y_i = k)[l(\mathbf{x}_i, k; A_\mathbf{s}) - l(\mathbf{x}_j, k; A_\mathbf{s})]$$

$$+ \sum_{k \in [C]} p_{y_i \sim \eta_k(\mathbf{x}_j)}(y_i = k) l(\mathbf{x}_j, k; A_\mathbf{s})$$

$$\leq \delta \lambda^l$$

where last step is coming from the fact that the trained classifier assumed to have 0 loss over training points. If we combine them,

$$E_{y_i \sim \eta(\mathbf{x}_i)}[l(\mathbf{x}_i, y_i, A_\mathbf{s})] \leq \delta(\lambda^l + \lambda^\mu L C)$$

We further use the Hoeffding's Bound and conclude that with probability at least $1 - \gamma$,

$$\left| \frac{1}{n} \sum_{i \in [n]} l(\mathbf{x}_i, y_i; A_\mathbf{s}) - \frac{1}{|\mathbf{s}|} \sum_{j \in \mathbf{s}} l(\mathbf{x}_j, y_j; A_\mathbf{s}) \right| \leq \delta(\lambda^l + \lambda^\mu L C) + \sqrt{\frac{L^2 \log(1/\gamma)}{2n}}$$

□

