# OpenReview forum: "Active Learning for Convolutional Neural Networks: A Core-Set Approach"
_ICLR.cc/2018/Conference — Accept (Poster)_

### Official Review · AnonReviewer1 · 2017-11-26
**This is interesting work and the results are very nice. But, the proof for Lemma 1 seems incomplete to me, and some choices (such as choice of loss function) are not properly justified. Also, important references in active learning literature are missing.**

**Rating:** 7
**Confidence:** 4

**Review:**

After reading rebuttals from the authors: The authors have addressed all of my concerns. THe additional experiments are a good addition.

************************
The authors provide an algorithm-agnostic active learning algorithm for multi-class classification. The core technique is to construct a coreset of points whose labels inform the labels of other points.  The coreset construction requires one to construct a set of  points which can cover the entire dataset. While this is NP-hard problem in general, the greedy algorithm is 2-approximate. The authors use a variant of the greedy algorithm along with bisection search to solve a series of feasibility problems to obtain a good cover of the dataset each time.  This cover tells us which points are to be queried. The reason why choosing the cover is a good idea is because under suitable Lipschitz continuity assumption the generalization error can be controlled via an appropriate value of the covering radius in the data space.  The authors use the coreset construction with a CNN to demonstrate an active learning algorithm for multi-class classification.
The experimental results are convincing enough to show that it outperforms other active learning algorithms. However, I have a few major and minor comments.

Major comments:

1. The proof of Lemma 1 is incomplete. We need the Lipschitz constant of the loss function. The loss function is a function of the CNN function and the true label. The proof of lemma 1 only establishes the Lipschitz constant of the CNN function. Some more extra work is needed to derive the lipschitz constant of the loss function from the CNN function.

2. The statement of Prop 1 seems a bit confusing to me. the hypothsis says that the loss on the coreset = 0. But the equation in proposition 1 also includes the loss on the coreset. Why is this term included. Is this term not equal to 0?

3. Some important works are missing.  Especially works related to pool based active learning, and landmark results on labell complexity of agnostic active learning.
UPAL: Unbiased Pool based active learning by Ganti & Gray. http://proceedings.mlr.press/v22/ganti12/ganti12.pdf
Efficient active learning of half-spaces by Gonen et al. http://www.jmlr.org/papers/volume14/gonen13a/gonen13a.pdf
A bound on the label complexity of agnostic active learning. http://www.machinelearning.org/proceedings/icml2007/papers/375.pdf

4.  The authors use L_2 loss as their objective function. This is a bit of a weird choice given that they are dealing with multi-class classification and the output layer is a sigmoid layer, making it a natural fit to work with something like a cross-entropy loss function. I guess the theoretical results do not extend to cross-entropy loss, but the authors do not mention these points anywhere in the paper. For example, the ladder network, which is one of the networks used by the authors is a network that uses cross-entropy for training.

Minor-comment:
1. The feasibility program in (6) is an MILP. However, the way it is written it does not look like an MILP. It would have been great had the authors mentioned that u_j \in {0,1}.

2. The authors write on page 4, "Moreover, zero training error can be enforced by converting average loss into maximal loss". It is not clear to me what the authors mean here. For example, can I replace the average error in proposition 1, by maximal loss? Why can I do that? Why would that result in zero training error?

On the whole this is interesting work and the results are very nice. But, the proof for Lemma 1 seems incomplete to me, and some choices (such as choice of loss function) are unjustified. Also, important references in active learning literature are missing.

---

> ### Author Response · Authors · 2017-12-19
> **Addressing reviewer comments**
>
> We thank the reviewer for their time and effort spent providing feedback. We appreciate the encouraging comments. We address the concerns as follows:
>
> Major Points:
> 1) Proof of Lemma 1: The reviewer is right as the current proof of the Lemma 1 is incomplete. Proposition 1 only requires the loss function to be a Lipschitz continuous function of the x(input data point) for a fixed true label (y) and parameters (w). Current proof of the Lemma 1 indeed proves this more restrictive but sufficient statement. Hence, we re-stated the Lemma 1 correctly in the updated submission. We also added the final step (reverse triangle inequality) to the proof of the Lemma 1 for sake of completeness.
>
> 2)The statement of Prop 1: We stated the proposition in this form to be consistent with equation 3. We clarified this in the updated text
>
> 3)We updated the related work with the suggested references.
>
> 4) L_2 loss: We agree that this is unconventional as cross-entropy is the widely accepted choice of loss function for classification problems. We use L_2 loss for theoretical simplicity and used cross-entropy in the experiments. The experimental results suggest that our method is effective for cross-entropy loss as well. We explicitly discussed and clarified this choice in the updated submission.
>
> Minor Points:
> 1)MILP: We added the missing constraint -u_j \in {0,1}- in the updated submission
> 2)Maximal Loss: We agree that this is a confusing statement and we removed it in the updated submission. What we meant was although 0 training error on core-set is a hard assumption, it can be directly enforced by using the maximal loss while learning the model. However, we did not use any such trick.
>
> We hope that the updated version answer the reviewer's concerns. Moreover, we also added more details and included two additional baselines in the updated submission.

---

### Official Review · AnonReviewer3 · 2017-11-27
**Active learning for CNN**

**Rating:** 7
**Confidence:** 4

**Review:**

Active learning for deep learning is an interesting topic and there is few useful tool available in the literature. It is happy to see such paper in the field. This paper proposes a batch mode active learning algorithm for CNN as a core-set problem. The authors provide an upper bound of the core-set loss, which is the gap between the training loss on the whole set and the core-set. By minimizing this upper bound, the problem becomes a K-center problem which can be solved by using a greedy approximation method, 2-OPT. The experiments are performed on image classification problem (CIFAR, CALTECH, SVHN datasets), under either supervised setting or weakly-supervised setting. Results show that the proposed method outperforms the random sampling and uncertainty sampling by a large margin. Moreover, the authors show that 2-OPT can save tractable amount of time in practice with a small accuracy drop.

The proposed algorithm is new and writing is clear. However, the paper is not flawless. The proposed active learning framework is under ERM and cover-set, which are currently not supported by deep learning. To validate such theoretical result, a non-deep-learning model should be adopted. The ERM for active learning has been investigated in the literature, such as "Querying discriminative and representative samples for batch mode active learning" in KDD 2013, which also provided an upper bound loss of the batch mode active learning and seems applicable for the problem in this paper. Another interesting question is most of the competing algorithm is myoptic active learning algorithms. The comparison is not fair enough. The authors should provide more competing algorithms in batch mode active learning.

---

> ### Author Response · Authors · 2017-12-27
> **Minor Clarifications**
>
> We thank the reviewer for their time and effort spent providing feedback. We appreciate the encouraging comments. We address the concerns as follows:
>
> - ERM and Core-Set for Deep Learning: Recent results (Example 7 of Robustness and Generalization by Xu&Manor 2010) provides a generalization bound for neural networks. Hence, CNNs support ERM. Moreover, our upper bound on core-set is also provided directly for CNNs. In summary, we believe our theoretical analysis is valid for CNNs. We update the paper accordingly and discuss the ERM for deep learning in Section 4.1.
>
> - Wang et al., KDD 2013: We thank the reviewer for pointing out this very related paper we missed in the original submission. We updated our related work section accordingly. Moreover, we also compare with this paper as well as another clustering-based batch-mode active learning baseline in the updated version. Our experiments suggest that these baselines turns out to be not effective for CNNs. We believe this is largely due to the fact that (Wang et al., KDD 2013) heavily uses uncertainty information and treating soft-max probabilities as uncertainty is misleading in general.

---

### Official Review · AnonReviewer2 · 2017-11-28
**An interesting work with both theoretical and empirical results**

**Rating:** 7
**Confidence:** 3

**Review:**

This paper studies active learning for convolutional neural networks. Authors formulate the active learning problem as core-set selection and present a novel strategy.

Experiments are performed on three datasets to validate the effectiveness of the proposed method comparing with some baselines.

Theoretical analysis is presented to show the performance of any selected subset using the geometry of the data points.

Authors are suggested to perform experiments on more datasets to make the results more convincing.

The initialization of the CNN model is not clearly introduced, which however, may affect the performance significantly.

---

> ### Author Response · Authors · 2018-01-05
> **Initialization details and the source code**
>
> We thank the reviewer for their time and effort spent providing feedback. We appreciate the encouraging comments. We revised the paper with the initialization details of the CNNs. Moreover, we are also planning to release the source code of our method as well as all the experiments for full reproducibility.

---

### Public Comment · (anonymous) · 2017-11-27
**the experiment is not solid; should compare with more baselines; the results is not convincing**

(1). The author considered a batch selection setting of active learning. In the field of batch mode active learning, there are a lot of strategies to overcome the information overlap problem in batch setting, such as clustering-based methods (Demir, B.; Persello, C.; and Bruzzone, L. 2011. Batch-mode active-learning methods for the interactive classification of remote sensing images.) and combining uncertainty sampling and diversity (i.e. Yang, Y.; Ma, Z.; Nie, F.; Chang, X.; and Hauptmann, A. G. 2015. Multi-class active learning by uncertainty sampling with diversity maximization. IJCV). However, the baseline methods the author considered is the simplest and stupid way: selecting the "top K" samples on its own criterion. Obviously, this approach cannot work well. The author should compare with some  smarter batch mode active learning methods, rather than the "top K" strategies.

(2). This paper argued "Since we have no labels available, we perform the core-set selection without using the labels." There already exists some active learning methods without using label information, such as Optimal experimental design (e.g., Yu, K.; Bi, J.; and Tresp, V. 2006. Active learning via transductive experimental design. ICML). It is quite similar to the proposed core-set idea. The author should compare with this kind of methods. The proposed core-set idea is also very similar to the K-medoids baseline (in
A Meta-Learning Approach to One-Step Active Learning, arXiv:1706.08334): the examples to label are selected following a k-medoid clustering technique, where we label each example if it is a centroid of a cluster. This paper should also compare with this baseline.

(3) The author argued in Section 5 "Our results suggest that both oracle uncertainty information and Bayesian estimation of uncertainty is helpful". We cannot find that the two uncertainty information is helpful. Because these methods perform comparable or even worse than random sampling in Figure 3. I think this statement is not correct.

(4) I would suggest to compare with the following baseline: using CNN extract the features, following by K-means or K-medoids clustering, where K is the batch size. Note that you need repeat the K-means several times (i.e. 50 times) to get a good initialization. I would expect that this baseline presents a very promising result. I would like to see the performance comparison of the proposed core-set method and this baseline.

---

> ### Author Response · Authors · 2017-12-19
> **Implemented the suggested baseline and added the suggested references**
>
> We thank for the comment and the pointers to the related literature we missed in the original submission. We answer the comments as;
>
> 1,2 and 4) We added all suggested work in the related work. However, direct comparison with them seems infeasible. The work by Yang et al. is not tractable for CNNs since it requires inversion of a very large matrix (number of data points by number of datapoints). The work by Yu et al. only applies to linear regression case with Gaussian noise assumption. Work by Demir et al. is indeed partially applicable to CNNs and we implemented and experimented with it. We also implemented the suggested k-Median baseline. Since both methods performed similarly, we only include k-Median in the experimental results in the updated submission. We also added another batch active learning baseline (Zheng Wang and Jieping Ye. Querying discriminative and representative samples for batch mode
> active learning. ACM TKDD 2015.)
>
> k-Median method turns out to be NOT effective for batch-mode deep learning. This is rather intuitive since the pool size is significantly larger than the query budget in all our experiments. For such a case, cluster centers happen to be around the modes of the data distribution. However, the important requirement of the query selection method is sampling the tails of the distribution instead.  Moreover, the neighborhood of such cluster centers are likely to be sampled in the initial labelled pool since they are near modes. Hence, clustering based methods fail for very large pool and a rather small budget case. Distribution matching based batch active learning baseline (Wang&Ye 2015) also fails since it heavily uses uncertainty information and treating soft-max probabilities as confidence values is misleading. We updated the text accordingly.
>
> 3) By that statement we meant, both oracle uncertainty and Bayesian estimation of uncertainty performs better than empirical uncertainty baseline. Hence, they are helpful improvements over empirical uncertainty baseline but still not effective when compared with random sampling. We clarified the claim in the updated version.

---

> > ### Public Comment · (anonymous) · 2018-01-06
> > **uncertainly information is not that bad; the reason why K-median failed is not convinced**
> >
> > Thank you for your answers!
> >
> > Since  K-means and k-Median are highly sensitive to initialization, I wonder what is the number of times to repeat clustering using new initial cluster centroid positions? What is this number used in your experiments? We find that the key of obtain good performance for k-means based batch mode active learning is repeating clustering multiple times with different random initialization. Your argued that the pool size is significantly larger than the query budget. This is exactly why you need repeat K median clustering many times to get good performance. In addition, in Demir et al. work, they use uncertainly information to select the sample from each cluster.
> >
> > Yu's work, (Active learning via transductive experimental design. ICML, 2006), can also be applied to non-linear case. See (Non-greedy Active Learning for Text Categorization using Convex Transductive Experimental Design, 2008, ICML). Yu also provide a sequential solution, which is easy to implement even for large scale datasets. It would be interesting if you can compare with Yu's work. (Because experimental design is mainly about the representativeness, which is quite similar to your proposed core-set idea).
> >
> > Another good baseline is Chakraborty, S., 2015. Active batch selection via convex relaxations with guaranteed solution bounds. TPAMI. It provides a theoretical guarantee for batch mode active learning.
> >
> > Finally, the main point of this paper is that uncertainty information is not effective for CNNs. However, according to this paper, I would only say that "top K" strategy based uncertainty information is not effective for batch mode active learning when applied to CNNs. The most straightforward to use uncertainty information for batch setting is the approach in (Patra, S., and Bruzzone, L. 2012a. A batch-mode active learning technique based on multiple uncertainty for svm classifier.) Patra's approach can make use of uncertainty information for batch setting, even in the case of CNNs. The only issue is that you need repeat K-means many times.

---

> > > ### Author Response · Authors · 2018-01-10
> > > **We used 100 random initialization for k-Median and will add the suggested baselines in the final version**
> > >
> > > We use 100 random initialization in the k-Median experiment in the paper. We solve each of the k-Median problems and choose the clustering having the minimum k-Median loss. Hence, we believe the results are not due to the random initialization. Moreover, we also qualitatively explain why we think k-Median heuristic fails for large-scale problems in the paper. We believe it fails to sample tails of the distribution since cluster centers are around modes of the distribution.
> > >
> > > We already added a new batch-mode baseline ("Querying discriminative and representative samples for batch mode active learning" in KDD 2013) which is the state-of-the-art uncertainty based batch method. Moreover, for the sake of completeness, we will further compare with the suggested baselines in the final version of the paper.
> > >
> > > Finally, we disagree that the main point of the paper is "...uncertainty information is not effective for CNNs...". The main point of our paper is defining the active learning as core-set selection and theoretically solving this core-set selection problem for CNNs.

---

### Public Comment · (anonymous) · 2017-12-13
**Coreset for k-center**

The authors construct coreset for k-center and use it for deep learning.
On the positive side it is great to see such new applications for core-sets.

On the negative side:
1) Novelty: the idea of using coreset for k-center and its guarantees for problems other than k-center (but not deep learning) was already suggested e.g. in: Visual Precis Generation using Coresets, Dan Feldman, Rohan Paul, Daniela Rus and Paul Newman.
IEEE International Conference on Robotics and Automation (ICRA) 2014

2) The "Active learning" approach is simply the classing hitting set approach for computing k-center.
See e.g.: https://arxiv.org/abs/1102.1472 by Karthekeyan Chandrasekaran, Richard Karp, Erick Moreno-Centeno, Santosh Vempala

3) Coreset for k-center has size exponential in d and in deep learning d is extremely large. Coreset of size independent or d might be more useful for these experiments e.g.:
Efficient Frequent Directions Algorithm for Sparse Matrices,  by Mina Ghashami, Edo Liberty, Jeff M. Phillips https://arxiv.org/abs/1602.00412

k-Means for Streaming and Distributed Big Sparse Data by Artem Barger, and Dan Feldman
https://arxiv.org/abs/1511.08990

Turning Big Data into Tiny Data: Constant-size Coresets for k-means, PCA and Projective Clustering,
Dan Feldman, Melanie Schmidt and Christian Sohler.
Proc. 24th Annu. ACM  Symp. on Discrete Algorithms (SODA) 2013

4) Experiments: the classic benchmarks are with much larger datasets such as Ciphar10/100 etc, with much larger networks.

---

> ### Author Response · Authors · 2017-12-27
> **Clarification**
>
> We thank for the comment. We added the suggested references in the related work section.
>
> We would like to clarify the following mis-understandings:
>
> 1-3) We are NOT addressing the coreset for k-Center problem. We are also NOT constructing coreset for k-Center in our algorithm. The main problem we address in this paper is the coreset for CNNs. Our theoretical study shows that the construction of the coreset for CNNs requires solving the k-Center problem as a sub-routine. Hence, we (approximately) solve the k-Center problem. Our solution of k-Center does not use any coreset and it is based on a mixed integer program (MIP).
>
> 4) We already performed experiments on Cifar10/100 in the original version. We also use a very large network (VGG16) in all of our experiments.

---

### Author Response · Authors · 2018-01-11
**Summary of Changes in the Updated Paper**

We thank all the reviewers and public commentators for their time and effort spend on our paper.

The main changes are as follows:
- Fixed the Lemma1: Proof of our Lemma 1 was incomplete as noted by R1. We fixed the statement and proof of the Lemma.
- New Baselines: We added two new baselines. One is a clustering based baseline as applying k-Median, and the other one is the state-of-the-art batch-mode active learning algorithm ("Querying discriminative and representative samples for batch mode active learning" in KDD 2013). Our method significantly outperforms both baselines, and we explain the details in the paper.
- Initialization: We added details on initialization of the neural network weights.
- Related Work: We added all the missing references suggested by reviewers and public commentators.
- We updated the main text to clarify confusing points and added the missing details.

---

### Public Comment · ~Yichong_Xu1 · 2018-05-17
**Comments and questions**

Thanks for the paper! It is a strong paper illustrating the strength of active learning for neural networks.
1. If I understand it correctly, the set-cover based strategy stated in this paper does not depend on the acquired labels, since it tries to find a set cover on all samples x_i. I believe this setting is called the "fixed design" setting more often. For example,
Rajkumar, Arun, and Shivani Agarwal. "When can we rank well from comparisons of O (n\log (n)) non-actively chosen pairs?." Conference on Learning Theory. 2016.
Shah, Nihar B., et al. "Estimation from pairwise comparisons: Sharp minimax bounds with topology dependence." The Journal of Machine Learning Research 17.1 (2016): 2049-2095.
I think it is at least worth pointing out in the paper.
2. As a result, in Theorem 1 it should be the set cover of {x_i}_{i=1}^n instead of {x_i,y_i}_{i=1}^n.
3. For experiments - For the semi-supervised learning setting,
Qiao, Siyuan, et al. "Deep Co-Training for Semi-Supervised Image Recognition." arXiv preprint arXiv:1803.05984 (2018).
reports error rates of 3.29% on SVHN using 1000(~1.5%) labels, and 8.35% on CIFAR-10 using 4000 (~6.7%) labels. Results reported in this paper seems much lower than these numbers. Your baseline method, Ladder network also have 20.30% error rate using 4000 labels on CIFAR-10, whereas your number is around 70% using 10% labels. In what setting will the active learning method be useful? For the fully-supervised setting, you also have already used the unlabeled samples when you select samples.

---

> ### Author Response · Authors · 2018-08-17
> **Thanks for the Comments**
>
> Thanks for the pointers. The camera ready for ICLR is over; but, we will update the Arxiv version with relevant papers.
>
> For supervised learning, we do not use the unlabelled points in learning the model, we only use the while selecting the points to query labels for.
>
> For the numbers, our method use the implementation and hyperparameters shared by Torch (http://torch.ch/blog/2015/07/30/cifar.html). Ladder network use a different architecture, so we guess that's the reason for the discrepancy. Regardless, all comparisons are using same architectures and hyper-parameters. We do not have any claims on SOTA on any of the dataset. We only claim our method out-performs other active learning methods using same architecture and hyperparameters.

---

### Decision · Program_Chairs · 2018-01-29
**ICLR 2018 Conference Acceptance Decision**

**Decision:**

Accept (Poster)

**Comment:**

The effectiveness of active learning techniques for training modern deep learning pipelines in a label efficient manner is certainly a very well motivated topic. The reviewers unanimously found the contributions of this paper to be of interest, particularly nice empirical gains over several natural baselines.